# Microbiome and Prostate Cancer: A Novel Target for Prevention and Treatment

**DOI:** 10.3390/ijms24021511

**Published:** 2023-01-12

**Authors:** Natasa Kustrimovic, Raffaella Bombelli, Denisa Baci, Lorenzo Mortara

**Affiliations:** 1Center for Translational Research on Autoimmune and Allergic Disease—CAAD, Università del Piemonte Orientale, 28100 Novara, Italy; 2Immunology and General Pathology Laboratory, Department of Biotechnology and Life Sciences, University of Insubria, 21100 Varese, Italy; 3Molecular Cardiology Laboratory, IRCCS-Policlinico San Donato, San Donato Milanese, 20097 Milan, Italy

**Keywords:** prostate cancer, microbiome, inflammation, immunotherapy

## Abstract

Growing evidence of the microbiome’s role in human health and disease has emerged since the creation of the Human Microbiome Project. Recent studies suggest that alterations in microbiota composition (dysbiosis) may play an essential role in the occurrence, development, and prognosis of prostate cancer (PCa), which remains the second most frequent male malignancy worldwide. Current advances in biological technologies, such as high-throughput sequencing, transcriptomics, and metabolomics, have enabled research on the gut, urinary, and intra-prostate microbiome signature and the correlation with local and systemic inflammation, host immunity response, and PCa progression. Several microbial species and their metabolites facilitate PCa insurgence through genotoxin-mediated mutagenesis or by driving tumor-promoting inflammation and dysfunctional immunosurveillance. However, the impact of the microbiome on PCa development, progression, and response to treatment is complex and needs to be fully understood. This review addresses the current knowledge on the host–microbe interaction and the risk of PCa, providing novel insights into the intraprostatic, gut, and urinary microbiome mechanisms leading to PCa carcinogenesis and treatment response. In this paper, we provide a detailed overview of diet changes, gut microbiome, and emerging therapeutic approaches related to the microbiome and PCa. Further investigation on the prostate-related microbiome and large-scale clinical trials testing the efficacy of microbiota modulation approaches may improve patient outcomes while fulfilling the literature gap of microbial–immune–cancer-cell mechanistic interactions.

## 1. Introduction

To date, among the male population, prostate cancer (PCa) represents the most frequently diagnosed non-skin cancer. Furthermore, it is qualified as a leading cause of cancer-related deaths worldwide. This condition poses a significant health concern in the future due to the gradual aging of the population [1].

PCa is a multifactorial and complex disease involving numerous genetic factors as well as environmental and physiologic factors [2]. Various factors have been associated and correlated to PCa, such as family history, age, diet, ethnicity, and viral and bacterial infections [3]. In addition, during the last decade, numerous studies have suggested the crucial role of the innate and adaptive immune system and environment [4].

The available treatments for PCa are numerous and can be more (prostatectomy or radiotherapy) or less (pharmacological treatment) radical. Treatments are associated with high rates of cure; nevertheless, a large proportion of PCa patients experience disease relapse within 10 years [5]. It is important to underline that 99% of patients with primary tumor diagnosis are characterized by a 5-year survival, while only 22% of subjects with metastatic disease, with bone as a primary site for dissemination, experience a 5-year survival [6]. PCa is portrayed as a tumor with considerable intratumor heterogeneity that has a great impact on the surrounding microenvironment and response to the therapy [7]. Even though conventional treatments, such as prostatectomy, chemotherapy, radiation, and androgen deprivation therapy (ADT), can improve overall survival rates in patients with metastatic PCa, the 5-year survival rate remains about 30% [8].

In the past two decades, our understanding of the factors that cause cancer and how they affect its development has evolved considerably [9]. In the case of PCa, the last decade has brought significant improvement in incidence and mortality worldwide that can be attributed to improved diagnostics, such as early PCa detection and therapeutic procedures. Nevertheless, despite worldwide efforts to find early diagnostic tools and new treatment strategies, the disease is still incurable in advanced stages, and that is precisely the reason why we need to improve our understanding of PCa development and progression (including premalignant lesions) and search for the clues in less evident sites.

Recent years have brought to intense research attention the direct or indirect relationship between cancer and the specific microflora of different cancer types, including PCa [10]. It has been shown that the microbiome can influence the development of cancer and response to therapies in two ways: directly exhibiting its effect on tumors or indirectly involving immune modulation, metabolic changes, and epithelial damage [11]. It has been revealed that specific microflora can slow down tumor growth [12]. Therefore, understanding the microbiome’s effects on cancer is critical to define potential therapeutic approaches [13,14].

The human microbiome is composed of bacteria, bacteriophages, viruses, fungi, and protozoa that are located in the epithelial surfaces of several areas of the body, such as the genitourinary tract, the skin, the oral cavity, and the gastrointestinal tract. All these body areas show notable interindividual differences in terms of microbial composition, hence representing a unique entity [15].

The microbiome affects several physiologic functions, such as inflammation, metabolism, hematopoiesis, as well as cognitive abilities [16].

In cases of extreme environmental change, the microbiome can slip into a state of dysbiosis that can further lead to the promotion of inflammatory diseases and cancer [17].

Nonetheless, there is still a limited amount of knowledge about the relationship between prostatic cancer and gastrointestinal tract and genitourinary tract microbiome, in terms of the impact of the microbiome on disease occurrence, development, progression, as well as response to medical treatments and the development of various resistance mechanisms. In this review, we explore the potential influence that the gastrointestinal and genitourinary microbiome exerts on prostatic cancer development, how specific bacteria are implicated in this type of cancer, and finally, how the microbiomes of these two body districts impact PCa treatments (Figure 1).

In the modern era of increasing personalized oncology, a deeper understanding of these relationships appears mandatory, all dedicated to one scope, improving the clinical outcomes of patients suffering from prostatic cancer.

## 2. Microbiome, Inflammation, and Prostate Cancer

Homeostasis depends on the balanced symbiotic relationship between the host and its microbiome. Different potential stimuli, such as microbial infections, chemical irritations, diet, obesity, and physical traumas, can promote the status of chronic inflammation. A pathogenic shift in the microbial species’ composition of the intraprostatic and genitourinary tracts, or dysbiosis, could lead directly or indirectly to an inflammatory state predisposing to the loss of the epithelial barrier integrity [11,18,19]. In turn, epithelial damage triggers an immune system response leading to the recruitment of the inflammatory cells, oxidative stress and the consequent DNA damage, compensatory epithelial proliferation, establishing a feed-forward mechanism promoting prostatic intraepithelial neoplasia (Figure 1).

Bacterial-infection-induced chronic inflammation, concurrent with epithelial barrier disruption, might be a key driver of an inflammatory microenvironment. The prostate microbiome and bacterial infections are one of the potential stimuli that drive an inflammatory microenvironment of the prostate promoting carcinogenesis [19,20]. Several bacterial species are known to infect the prostate and cause bacterial prostatitis, an infection or inflammation of the prostate gland. Infections such as bacterial prostatitis frequently linked to *E. coli* or other species of *Enterobacteriaceae* are identified as the leading causes of prostate inflammation [21]. In this context, various epidemiological studies have linked prostatitis with prostate cancer risk [22].

Prostatic chronic inflammation is prevalent in the adult prostate and has been suggested as a risk factor for prostate cancer development [23]. A high prevalence of chronic inflammatory infiltrates has been detected in prostate histopathological examinations. Accordingly, DNA and RNA from bacteria, fungi, parasites, and viruses have been found in prostatectomy samples from men who suffer from benign prostatic hyperplasia (BPH) and PCa [24]. The detection of inflammatory cells in the prostate microenvironment in adult men indicates that inflammation is involved in these conditions [25].

Currently, no single pathogenic species have been implicated in PCa development. Instead, microbial dysbiosis in the prostatic tissue can contribute to prostate inflammation in relation to benign prostate conditions, such as BPH, as well as PCa progression and/or treatment outcome [26]. Studies investigating microbial signatures in benign and malignant prostate tissues have found some evidence to support inflammatory responses associated with PCa-specific microbiota [24,27,28]. Banerjee et al. established a microbiome signature for PCa and predicted that *Helicobacter cagA* sequences integrated within specific chromosomes of prostate tumor cells could affect the expression of several cellular genes associated with oncogenic processes [24]. Cavarretta et al. suggested that the pathological prostate is populated by specific microbial populations, among which *Propionibacterium* spp. (P. acnes, at present classified as *Cutibacterium acnes*) were the most abundant, while *Staphylococcus* spp. were more represented in the tumor tissues [27].

The microbiome can influence every stage of carcinogenesis directly or by affecting the response of the immune system and effects on therapy. Persistent cytokine and chemokine production associated with microbial inflammation can influence several biological processes, including immune cell infiltration and their activation/deactivation, angiogenesis, cancer cell proliferation, survival, metastasis, and therapeutic resistance.

*C. acnes* is the most prevalent microorganism isolated from prostatic tissue by several independent studies [29,30,31,32,33,34]. Its relevance as a cytokine and inflammation-inducing agent has been widely discussed [31,35,36,37]. *C. acnes* was reported to induce long-term NF-κB activation, suggesting the major reprogramming of host cell inflammatory signaling [37]. Long-term exposure to *C. acnes* altered cell proliferation and initiated cellular transformation [34]. Additionally, a strong multifaceted inflammatory response with increased levels of IL-6, which is associated with advanced metastases in PCa, has been reported [31].

A large proportion of the metabolic products in the blood that originates from the gut microbiome can provoke changes to immune function that may result in either tumor-promoting inflammation or enhanced anti-tumor immunity. There is a strong impact of microbial metabolic processes on cytokine production. TNFα and IFNγ production, two key factors in determining the inflammatory process, appear to be more strongly influenced by the microbiome [38].

Toll-like receptors (TLRs) are critical in sensing the microbiota, maintaining tolerance, or eliciting an immune response through the direct recognition of ligands derived from commensal microbiota or pathogenic microbes. By using a whole-transcriptome profiling approach, Salachan et al. found decreased species diversity and significant under-representation of *Staphylococcus saprophyticus* and *Vibrio parahaemolyticus*, as well as a significant over-abundance of *Shewanella* in malignant as compared to benign prostate tissue samples [28]. Interestingly, the authors suggested that increased *Shewanella genera* is associated with the downregulation of genes involved in TLR signaling pathway and decreased enrichment of dendritic cells. The downregulation of the TLR signaling pathway in the malignant tissue samples with a high *Shewanella* count could indicate a pathogen-associated decreased host immune response. Malignant samples were also enriched for M1 and M2 macrophages as compared to benign tissue samples, suggesting an ongoing dysregulated inflammatory response. Finally, *Microbacterium sp.* was found to be significantly over-abundant in pathologically advanced prostate cancer tissues [28].

Inflammatory stress and changes within the prostate microenvironment might contribute to genomic alterations leading to prostatic intraepithelial neoplasia and PCa progression. However, the relationship between inflammation, the microbiome, and PCa is yet to be elucidated.

## 3. The Genitourinary Microbiome and Prostatic Cancer

### 3.1. The Genitourinary Microbiome

Given the physiological function and anatomical position of the prostate gland, studies on the genitourinary tract microbiota are of major significance in research of PCa. Conversely, despite the overall present speculations on the possible association between the genitourinary tract microbiome and PCa, very few trials have been conducted to date on this topic.

As defined above, microbiome is the sum of all genomic information belonging to the resident microbiota that colonizes various anatomical niches of the body [39,40]. One such anatomical niche is genitourinary tract. For the purposes of this review, the term “genitourinary tract” refers to the organ system involved in the production, transport, storage, and excretion of urine, namely, the kidney, ureter, bladder, and urethra and organs involved in reproduction that may contribute to the microbial load, such as the penis, pubic skin surfaces, and perianal area. The composition of the genitourinary tract microbiome is very specific, and it is influenced by a number of elements, such as gender, age, sexual behaviour, and concomitant diseases [41,42]. It has been proven that the human genitourinary tract contains a variety of resident microbial communities [43], hence disapproving the traditional view of urine being sterile [44]. To this point, the presence of shared species in the urinary tract and those found in microbial populations of gastrointestinal tract, vagina, or even skin was well defined. Nevertheless, some recent studies have suggested that the urinary tract contains microbial populations that are different from those at other sites of the human body that are populated by microbiomes [45,46].

The genitourinary tract microbiota has been characterized and reproducibly measured. To date, more than 100 species from more than 50 genera are thought to reside in the human genitourinary tract [47,48,49]. The majority of microbiome species identified to date belong to five major phyla, *Firmicutes*, *Bacteroidetes*, *Actinobacteria*, *Fusobacteria*, and *Proteobacteria* [50], and commonly include the genera *Lactobacillus*, *Corynebacterium*, *Prevotella*, *Staphylococcus*, and *Streptococcus* [51]. The genera *Lactobacillus* and *Gardnerella* are predominant in the female microbiota, whereas the male microbiota presents a higher percentage of *Corynebacterium*, *Staphylococcus*, and *Streptococcus* [46,52]. Nevertheless, interindividual variability is profoundly present resulting in still ill-defined members of a core genitourinary tract microbiome.

One of possible causes for the increased risk of PCa is inflammation due to the infection of the urinary tract (UTi) [53]. In addition, certain microorganisms that have been identified as key players in those infections have also been implicated in carcinogenesis [54]. The inflammation of the prostate, resulting from exposure to microbial agents, stimulates the production of inflammatory cytokines and reactive oxygen species, leading to increased cellular proliferation and, possibly, to carcinogenesis [55]. Through a link between the infectious agents, inflammation, and the location of the prostate gland, a possible association between UTi and PCa is inevitable [56]. To date, most studies have focused on the role of prostatitis and sexually transmitted diseases in the development of PCa [57,58]. On the other hand, studies that are investigating the link between UTi and PC are few, often inconclusive, and contradictory. Urinary tract infection can be divided into lower (presence of microbial pathogens in the urethra or bladder) and in upper tract infections (presence of microbial pathogens in the ureter and pelvis of the kidney). The prostate is located in the pelvis, adjacent to the bladder, and surrounding segments of the urethra; hence, theoretically, lower urinary tract infections, such as urethritis and cystitis, may cause inflammation of the prostate and play a role in the development of PC. The mechanism underlying the association between lower UTi and PCa is the focus of ongoing research. A major pathogen involved in UTi is *Escherichia coli* and is assumed at present that this pathogen is responsible for nearly 80% of all UTi [59]. Nevertheless, *E. coli* has not yet been connected directly to carcinogenesis. Furthermore, pathogens that link PCa and UTi are found with the Gram-positive bacillus, *Propionibacterium acnes* and *Trichomonas vaginalis*, the most common non-viral sexually transmitted infection. It has been proposed that pathogens can cause the inflammatory response that may play an important role in the development of PCa [34,60]. It is interesting to emphasize that pathogens such as *Chlamydia trachomatis* and human papillomavirus (HPV), which are extremely common in sexually transmitted infections, have almost no association with an increased risk of developing PCa [61,62]. Pelucchi and colleagues found that cystitis increased the risk of PCa occurrence by 76% [63]. Fan et al. found a possible role for urethritis and cystitis in the development of PCa. They found that urethritis had the most significant increase in PCa risk (95% CI: 1.26–2.34). However, this study included only Taiwanese men as participants and, hence, the findings may not be generalizable, even though the sample size was considerably large, namely, the authors enrolled 14,273 patients [56]. On the other hand, Boehm et al., in a similar study with a focus on urethritis, concluded that this UTi was not associated with PCa [64]. Lightfoot et al. found that urinary tract infection was not associated with PCa in a population-based case–control study with 760 cases and 1632 controls [65]. Russell et al. performed a population-based case–control study and did not find any association between UTi and the increased risk of developing PCa. The authors found that only subjects that had a UTi within the past year were found to have an increased risk of PCa diagnosis by 49% [66]. A potential explanation of association between UTi and PCa is that cystitis or urethritis may cause bacterial prostatitis or chronic and asymptomatic inflammation of the prostate, which can further lead to carcinogenesis, similarly to the mechanism by which sexually transmitted diseases can cause PCa [59]. Asymptomatic prostatic inflammation is common in adult males, and has been confirmed by the fact that inflammatory cells were found in the prostate biopsy, or leukocytes found in semen analysis from patients without a history of prostatitis [67]. Inflamed tissue produces active oxygen and nitrogen radicals that increase the risk of cancer by suppressing antitumor activity and stimulating carcinogenesis [68,69]. Reduced activity of glutathione S-transferase P1 (an enzyme that protects the genome from oxidative damage) leads to the formation of an inflammatory lesions in prostate, which is thought to be a precursor of PCa [70].

Alterations of the urinary microbiome affect the response to the anti-microbial treatment of urinary tract infections [71]. Genitourinary tract infections are the most common bacterial infections experienced in adults and, as such, impose a substantial medical burden to the health care system. Infections of the genitourinary tract affect predominantely women, but in addition to gender, the incidence of genitourinary tract infections also depends on age, sexual behaviour, and concomitant diseases. Infections of the genitourinary tract can be caused by a variety of viruses, bacteria, and fungi, but the most common pathogens have been identified as uropathogenic *Escherichia coli*, *Klebsiella pneumoniae*, *Enterococcus faecalis*, and *Proteus mirabilis* [72]. Usually, the treatment of choice in the case of genitourinary tract infections is antibiotic therapies, and as simple this may seem, unfortunately, the overuse of antibiotic treatment frequently leads to the antibiotic resistance and/or allergy, which will further lead to the recurrence of the infections and/or potentially dangerous complications, such as pyelonephritis and urosepsis [73,74].

Several factors are involved in shaping the microbiome milieu and, among them, nutrient availability, osmolarity, pH levels, oxygen tension, adhesion sites, and immune interaction are listed as the most important. In the human urine are present numerous soluble elements, prevalently electrolytes and osmolytes but also amino acids and carbohydrates. To date, more than 2500 compounds have been detected in the urine [75].

The constant flux of new urine is likely responsible for supplying nutrients to resident microbes [76]. Furthermore, specific pH that can vary from 5 to 8 in healthy individuals [77] as well as oxygen availability, without a doubt, play a role in shaping the microbiome of the genitourinary tract [78].

### 3.2. Prostatic Cancer Microbiome

A potential state-of-the-art perspective could be positioned regarding the study of the microbial composition in PCa tissues. Nonetheless, to date, few trials have been conducted, and it remains unclear if there is a specific “prostate microbiome” or not [79,80]. Most of the studies conducted to date present several limitations, prevalently due to false positive results related to contamination. To date, the available studies suggest that the composition of the prostate microflora is similar to that of the urethra [27,44].

Hochreiter et al. performed one of the first prostate microbiomes collecting prostate tissues from 9 patients (7 subjected to radical prostatectomy and from 2 were obtained as simple prostatectomy specimens) and 18 healthy controls (HC) [81]. The authors performed 16S rRNA gene PCR in order to identify the presence/absence of bacteria. The obtained results showed no bacteria DNA was present in the HC (6 bacteria/25 mg of prostate tissue). On the other hand, the specimens obtained from patients diagnosed with PCa revealed the substantial presence of bacteria; nevertheless, the authors did not perform sequencing to identify the species. Even though this study had its imperfections and was not completed by identifications of the bacterial strains present in PCa, it was an early study that has paved the way towards the identification of the microbiome and its potential role in the pathology of PCa.

A very radical study with converse conclusions was performed in which radical prostatectomy tissue core samples were evaluated; 16S rDNA gene sequencing was performed and bacterial DNA was found in prostate tissues, but when this was compared to core samples, the biopsies were negative [82]. Furthermore, no significant correlation was observed between the presence of specific species of bacteria and histologic evidence of chronic or acute inflammation. On the other hand, focal regions presented several bacteria that are frequently observed in cases of infections of the urinary tract, such as *Escherichia* spp., *Enterococci* spp., and *Pseudomonas* spp. In addition, some other bacteria were found that are usually present in the urethral flora in physiological conditions, such as *Streptococcus* spp., *Acinetobacter* spp., and *Actinomyces* spp. The results led the authors to hypothesize that a prostatic microbiome may not exist, but that their finding is merely the result of remnant bacterial DNA that has been “fossilized” in the prostate tissue [82].

Several years later, a study performed by Yow and colleagues included 20 patients with aggressive PCa. Total RNA sequencing and 16S rDNA next-generation sequencing were used as a method to establish the presence of bacterial DNA in tumor areas as well as benign regions to areas with PCa. The authors confirmed the presence and prevalence of the *Enterobacteriaceae* family, with *Escherichia* and *Propionibacterium acnes* being the most numerous [79].

Cavarretta and colleagues noted that C. acnes, the most represented species of bacteria equally present in all tissues, is highlighted in patients with PCa [27]. The involvement in the pro-inflammatory pathway of P. acnes has been confirmed within prostate tissue in murine models, suggesting a potential involvement in the development of PCa [36,83]. Furthermore, the authors did note a larger proportion of *Streptococcaceae* in non-tumor tissues and a greater proportion of *Staphylococcaceae* in tumor tissues [27]. It is hypothesized that the exclusive presence of *Streptococcus* in non-tumor tissues may indicate a normal microbiome of healthy prostatic tissue. Nevertheless, one must bear in mind that both *Streptococcus* and *Staphylococcus* spp. are among the most common bacteria on human skin and, as such, are very frequently responsible for contaminations in laboratory analysis [84].

In a more recent study, Feng et al. [80] analysed the tissue samples of 65 patients who underwent radical prostatectomy. The authors identified *Pseudomonas*, *Escherichia*, *Acinetobacter*, and *Cutibacterium* as the most abundant bacteria present in the examined tissues. Conversely, there was no differences with the adjacent benign tissues [80]. In the same year, Banerjee et al. evaluated the presence of pathogens in tissue samples collected from 50 radical prostatectomy patients and 15 BPH patients who underwent transurethral resection of the prostate, using the pan-pathogen microarray metagenomics analysis (PathoChip) [24]. The authors identified a well-defined pathogenic microbiome in PCa patients. *Proteobacteria*, *Firmicutes*, *Actinobacteria*, and *Bacteroides* were the most frequently observed phyla, with no differences in terms of microbiota signatures between samples obtainedfrom the PCa patients and men without PCa. The most significant finding was the detection of *Helicobacteri pylori* in more than 90% of PCa specimens, further confirming *H. pylori*-cytotoxin-associated gene A (CagA) gene integration into the prostatic tumor DNA. The CagA gene is the virulence factor of *H. pylori* that has a known association with gastric cancer development through the activation of proto-oncogenes and the inactivation of tumor suppressor genes [85]. In addition, the authors noted the presence of several tumorigenic viruses, such as human cytomegalovirus, human HPV 16, and HPV 18, with these three agents accounting for 41% of all the viruses isolated [24].

Linked to the previous finding of the potential involvement of viruses in PCa pathology, it is noteworthy to mention the study performed by Miyake and colleagues in 2019 with the intention of evaluating the presence of agents involved in sexually transmitted infections and their connection to PCa. The authors obtained samples from 33 BPH and 45 PCa patients and tested them for the presence of several infectious agents, including HPV 16, HPV 18, *Ureaplasma urealyticum*, *Mycoplasma genitalium*, *Neisseria gonorrhoeae*, *Mycoplasma hyorhinis*, and *Chlamydia trachomatis*. Conversely, only *Mycoplasma genitalium* was associated with a higher Gleason score and PCa development [86].

A meta-analysis from 2009, which included thousands of patients with PCa and controls from 29 case–control studies, showed a significant association between carcinogenesis risk and infection history of any sexually transmitted infection, including *Mycoplasma genitalium* and HPV [87]. The role of inflammation as a key player in the initiation and progression of PCa was placed under intensive research. It was confirmed that inflammation stress coerces prostate carcinogenesis through excessive reactive oxygen species, epigenetic alterations, and subsequent mutagenesis [19]. In addition, chronic inflammation can be generated by the constant and intense exposure of the numerous microorganisms in the prostate through the urethra.

To date, several hypotheses have been formulated, including the concomitant “action” of several different microbes contributing to the increased risk of PCa, but still there are very few studies directly dealing with the potential role of genitourinary microbiome and PCa.

In 2016, one of the first studies was published in which the authors documented well the involvement of pro-inflammatory bacteria, such as *Escherichia coli*, *Streptococcus anginosusi*, and *Propionibacterium acnesi*, in acute and chronic prostatitis, which, in turn, may result in hyperplasia and a higher risk of PCa development [88]. Namely, lipopolysaccharide (LPS), the main component of the cell wall of bacteria such as *Escherichia coli* and *Neisseria gonorrhoeae*, is a bacterial endotoxin that is released after bacterial cleavage. Numerous genes that are included in cell proliferation, differentiation, and apoptosis are abnormally overexpressed when stimulated with LPS, which can lead to the epithelial–mesenchymal transition (EMT) of prostate cells [89]. Not only endotoxins, but also exotoxines can promote the progression of PCa. On the other hand, some bacterial toxins show anti-tumor properties, such as alpha toxins [90] and enterotoxins [91]. Enterotoxin expressed by *Staphylococcus aureus* can induce the apoptosis of PC3 cells through changes in the expression of lncRNAs, including Gas5, PCA3m and NEAT1 genes [92]. Botulinum toxin A can induce phospholipase A2 (PLA2) phosphorylation, leading to the inhibition of the growth and proliferation of PCa cells [93]. Thus, it can be seen that bacterial toxins have a great potential in the treatment of cancer.

Shrestha and colleagues performed a study in which they evaluated urine by using 16S rRNA gene amplicon sequencing. Samples were obtained from 135 men prior to prostate biopsy [49]. The results showed diverse bacterial populations present in the urine and also indicated a potential role of pro-inflammatory bacteria involved in a subset of PCa patients (e.g., *Actinobaculum schaalii*, *Anaerococcus lactolyticus*, *Anaerococcus obesiensis*, *Streptococcus anginosus*, *Propionimicrobium lymphophilum*, and *Varibaculum cambriense*). Nevertheless, there was no substantial difference between benign and cancerous samples [49]. However, the results of Shrestha et al. regarding the role of pro-inflammatory bacteria produced the hypothesis that pro-inflammatory bacteria may influence inflammation, urine reflux, and PCa [19].

Alanee and colleagues performed a study in which they examined not only the gut microbiota but also the urinary microbiota of 30 patients undergoing transrectal prostate biopsy. The authors strongly concluded that patients with PCa showed a high prevalence of *Bacteroides* and *Streptococcus* and a low prevalence of *Acinetobacter*, *Lactobacilli*, and *Faecalibacterium* compared to patients with BPH [94]. Nevertheless, the conclusion of this study needs to be taken cautiously considering the limited sample size and statistical power.

Having in mind the hypothesis that prostate microflora potentially originates from the urethra and intestinal tract, one can easily deduce the potential role that drugs and diet may have on the prostate microenvironment, as some previous studies have shown that some drugs or diets can alter the structure of urethral and gut bacteria [95]. The microbiota in the human body is highly susceptible to antibiotic therapy, which usually leads to its disbalance activating a variety of stress mechanisms, including genomic mutations/modifications and the production of enzymes to degrade antibiotics. Obviously, different drugs have different effects on prostate tissue. For example, in a study of quinolones, it was shown that the ability of the drug to penetrate the prostate tissue was in the order of norfloxacin < fluidixacin < ciprofloxacin < ofloxacin < fleroxacin [96].

Further trials are highly awaited in order to gain more insights into the effective role of the urinary microbiome in the onset and progression of PCa.

## 4. Gut Microbiome, Diet, and Prostate Cancer

### 4.1. Diet and Prostate Cancer

In recent years, diet and related obesity have been extensively investigated as important factors in increasing the risk of PCa. It is well known that diet and other lifestyle factors are able to modify the gut microbiome and, hence, influence numerous processes in the body.

It is difficult to compare the consumption of individual nutrient factors and determine their effects on PCa, since numerous factors other than diet are at play. Therefore, it is, perhaps, better to speak about dietary patterns rather than individual components.

Typically, dietary patterns are divided into two distinctive types: “Western diet” (WD) and “Prudent diet” (PD). WD includes a high intake of red meat, frequently highly processed, high-fat dairy products, vast quantities of bread, potatoes, and overall high carbohydrate intake. On the other hand, PD is quite the opposite and it is characterized by higher intake of vegetables, fruits, legumes, and fish [97].

Obesity resulting from a high-fat diet (HFD) induces chronic systemic inflammation, via cytokines or chemokines secreted from adipocytes and may be involved in PCa progression through immune-system-related mechanisms, particularly those associated with the function of MDSC and macrophages [98,99]. A meta-analysis published in 2012 has shown that men with a higher body mass index (BMI) have a higher risk of PC [100]. HFD usually leads to obesity, which results in hyperinsulinemia and increased amounts of circulating bioactive insulin-like growth factor-1 (IGF-1) that has the ability to promote the development of many types of cancer.

Furthermore, HFD can lead to local inflammation in the prostate, resulting in increased levels of IL-6, suppressing tumor immunity and thus leading to the growth of PCa [101]. Several other studies have confirmed the important role of HFD in PCa progression, most likely involving inflammation specific for prostate tissue that involves proinflammatory cytokines and chemokines [102,103,104,105]. Additionally, HFD may impact the metabolism of sex hormones, potentially leading to the progression of PCa. Nevertheless, current dietary intervention studies have not revealed that variations in dietary composition have any long-term major effects on circulating sex hormone levels in men [106].

On the other hand, the consumption of high amounts of ω-3 fatty acid (unsaturated fatty acid) reduced prostate tumorigenesis by lowering estradiol, testosterone, and androgen receptor levels in transgenic mice [107]. In conclusion, it can be said that the saturated fat of animal origin can contribute to the progression of PCa, while unsaturated fatty acids, abundant in fish and vegetable oils, reduce the risk of PCa [108]. It has been shown that the high consumption of monosaccharides and/or disaccharides likely contributes to PCa progression through IGF-1-mediated inflammation [109]. In addition, vitamins A, D, and E have been investigated regarding their influence on PCa prevention and, even though most of the studies suggest protective roles regarding PC development, more studies are needed since there are numerous conflicting results [110].

Several studies have confirmed the increased risk of PCa development and progression in the case of following the WD in different parts of the world [111,112,113,114]. Only two cohort studies from western countries reported no association [115,116]. The influence of diet on PCa has been confirmed and has received a considerable amount of attention. Without any doubt, it has been confirmed that certain nutrients (saturated fat, carbohydrates, vitamins, polyphenols, etc.) are involved in PCa pathogenesis through various mechanisms that include inflammation, antioxidant effects, sex hormones, and alterations of the gut microbiome (Figure 2). On the other hand, the gut microbiome affects PCa insurgence and progression via metabolite release. A high-fat diet can disrupt the gut microbial composition and cause gut dysbiosis and the release of gut bacterial metabolites, such as short-chain fatty acids and phospholipids that enter systemic circulation affecting distant organs [108,117]. Interventions to improve dietary habits can improve the gut microbiome, thus preventing or delaying prostate cancer development. For instance, promoting the use of prebiotics and /or probiotics may promote beneficial gut bacteria that may lessen the risk of developing PCa [118]. Further studies to determine the relationship between diet, gut microbiome and PCa can help to discover potential targets for the prevention, screening, and treatment of PCa.

### 4.2. Gut Microbiome and Prostate Cancer

Several pieces of evidence have pointed to the fact that the gut microbiome and PCa are linked by multiple factors, both pro-tumoral and anti-tumoral (Figure 2). Previous research has demonstrated that PCa patients’ intestines have a higher relative abundance of *Bacteroides massiliensis* in comparison to patients with benign PCa or healthy subjects, while, on the other hand, *Faecalibacterium prausnitzii* is in lower relative abundance [119]. *Faecalibacterium prausnitzii* is responsible for the metabolization of acetic acid, which can be metabolized to butyric acid, the most prevalent short-chain fatty acid (SCFA) in the colon. Butyric acid possesses anti-tumor activities, mostly by inducing apoptosis and decreasing proliferation [120].

A study from Liss et al. showed that, using 16S rRNA sequencing to detect rectal swabs, *Streptococcus* and *Bacteroides* species were higher in men with PCa than the control subjects, and they speculated that gut bacteria may interfere on the risk of PCa [121].

Of note, Matsushita et al. discovered that PCa development by HFD can be inhibited by the administration of antibiotics in a Pten-knockout PCa mouse model [122]. They found that antibiotics have a profound effect on the gut microbiome, dramatically diminishing the expression of IGF-1 in tumor tissue and blood. IGF-1 could be also produced by PCa cells and is a key regulator of cancer growth through the MAPK and PI3K enzymes, and these signaling pathways are blocked by antibiotics. SCFAs (butyrate, acetate, proprionate, and isopropionate) represent the main metabolites of the gut microbiota, acting on the regulation of PCa growth through the production of IGF-1 inside the prostate tissue and at the systemic level. It has been shown that antibiotic administration counteracted the growth of *Rikenellaceae* and *Clostridiales* in mice gut microbiome, responsible for the high production of SCFAs in the mice gut conditioned by an HFD, while diminishing SCFAs in mouse feces. Furthermore, HFD-fed mice receiving antibiotics and SCFAs showed the loss of the inhibitory effect on IGF-1 as well as the loss of suppression of PCa growth [122].

It is of note that HFD causes a leaky gut, and this leads different metabolites and bacterial fragments to enter the host systemic circulation, causing, for example, endotoxemia. This phenomenon can thus orchestrate the inflammatory response, influencing the regulation of PCa growth [98,99,101].

Another interesting study supporting the existence and importance of the “gut–prostate axis” is that of Liu et al., who used a PCa mouse model [123]. They showed that, when fecal microbiota transplant (FMT) derived from men with castrate-resistant prostate cancer (CRPC) were transferred to the transgenic adenocarcinoma of the mice prostate (TRAMP), high levels of gut *Ruminococcus* was induced and concomitantly PCa growth increased, probably due to the increase in lysophosphatidylcholine acyltransferase 1 (LPCAT1) (since *Ruminococcus* is correlated with phospholipid metabolism), given that its upregulation has been described in several types of cancers and is associated with a poor prognosis [124,125,126]. Liu et al. showed elevated levels of LPCAT1 in CRPC FMT-treated mice as well as RAD51 and DNA-dependent protein kinase catalytic subunits in the mice prostate [123].

Another crucial work on the relevance of the “gut–prostate axis” in PCa development was performed by Pernigoni et al., in which they demonstrated that the gut microbiota in patients with CRPC or castrated mice can generate active androgens from their precursors [127]. The authors showed that the antibiotic treatment of mice depressed the gut microbiota and diminished circulating dehydroepiandrosterone (DHEA) and testosterone levels. Additionally, they found that the presence of the *Ruminococcus genus* was preponderant in the gut microbiota of CRPC patients, which correlated with a poor prognosis compared to hormone-sensitive PCa (HSPC) patients, while the presence of the *Prevotella stercorea* was associated with a good prognosis. Importantly, *Ruminococcus* was able to metabolize pregnenolone and hydroxypregnenolone to DHEA and testosterone [127].

More recently, Terrisse et al. revealed new possible mechanisms of interaction between the gut microbiota and the immune system during androgen deprivation therapy (ADT) in preclinical mouse models and in CRPC and HSPC humans. The authors showed that PCa and ADT treatments are able to influence the function of the immune system in opposite manners. Furthermore, the authors claimed that thymus-dependent T cells are involved in the control of PCa progression since the depletion of CD4+ and CD8+ T cells resulted in the partial reduction in tumor growth control by therapy and a faster time of progression from HSPC to CRCP [128]. It has been reported that ADT is able to enriche gut bacteria *Akkermansia muciniphila* [129] and this was further confirmed by Terrisse et al., who showed that the depletion of beneficial bacteria, including *Akkermansia muciniphila* and *Lachnospiraceae*, was reversed by the effects of ADT. Furthermore, Terrisse et al. showed that PCa altered the gut microbiota (with loss of protective bacteria such as *Akkermansia muciniphila* and *Lachnospiraceae* spp.), and that these PCa-dependent microflora changes could have been reversed by multiple ways in a mice model: through cohousing PCa-bearing mice with cancer-free mice, using FMT from human healthy subjects, and the oral administration of live *Akkermansia*. Any of these methods was able to reduce PCa growth and/or improve the efficacy of ADT and delay progression from HSPC to CRCP. They also found that, when a combination of three high-spectrum antibiotics (ampicillin, colistin, and streptomycin) was used in mouse models, the anticancer effect of ADT was greatly reduced, corroborating the notion that specific gut bacteria, in particular those of alpha diversity as well as a high level of *Akkermansia muciniphila*, are associated with effective therapy [128]. This latter effect had similarities to the beneficial results obtained using oral anti-androgens in PCa patients with an abundance of *Akkermansia* [19,129].

## 5. Bacterial Immunotherapy for Prostate Cancer

When it comes to the bacterial immunotherapy and its employment in cancer treatments, most of the current research is limited to the gut bacteria, and there are very few studies that deal with PCa bacteria; nevertheless, prostate bacteria are a requisite in PCa immunotherapy. The use of microorganisms to prevent and treat PCa may be a tempting treatment strategy into the near future. Specific microorganisms can deliver exogenous genes to the cells of PCa or the metabolites as well, and can interact with essential processes, such as proliferation and/or apoptosis.

Currently, the microorganisms that can potentially become targeted therapies are mainly nonpathogenic bacteria and viruses, especially for anaerobes or facultative anaerobes, since the tumor microenvironment is usually accompanied by hypoxia. It was discovered that *Escherichia coli* (facultative anaerobe) could specifically produce TNF-α in mouse tumors [130]. It has been shown in numerous studies that TNF-α can induce tumor cell apoptosis; nevertheless, this kind of treatment is not feasible due to heavy systemic side effects [131].

*Salmonella typhimurium* (anaerobic bacteria) can induce the death of PC-3, LNCaP, and DU-145 PCa cells through different mechanisms [132].

*Serratia marcescens* (facultative anaerobes) can inhibit the growth of PCa cells through the downregulation of IAP family inhibitors, such as XIAP, cIAP-1, and cIAP-2, and the activation of caspase-9 and caspase-3 [133].

Based on previous research, it can be deduced that a future research trend in the treatment of PCa should be based on the exploration of the etiology and mechanism of PCa in the field of microbiology.

Microbial immunotherapy and targeted therapy can easily overcome the limitations of traditional therapy and it should be based on the in-depth study of the microbiome of the gut, genitourinary system, and PCa.

In microbial immunotherapy, the most evident criticism is potential biosafety problems in the use of microorganisms, especially recombinant viruses or bacteria that are widespread in laboratories. In clinical trials, it is often accompanied by some side effects. However, based on the already available therapies such as numerous vaccines, it is undeniable that these biosafety problems will be solved in the near future.

## 6. Developing Therapeutic Applications Targeting the Microbiome

Several compelling examples have shown that the gut and genitourinary microbiota can profoundly influence PCa therapy, such as ADT, oral androgen target therapy (ATT), and checkpoint inhibitors immunotherapy (ICI).

Recent human studies reported that the presence of certain types of bacteria, including *Ruminococcaceae*, *Bifidobacteriaceae*, and *Akkermansia muciniphila*, are associated with a response to anti-PD-1 immunotherapy.

Fecal microbiota transplant from human donors (responders to anti-PD-1 immunotherapy) into germ-free mouse (allograft tumor models) was able to reverse anti-tumor efficacy of anti-PD-1 immunotherapy [134,135]. These studies indicated that the genitourinary microbiome, with its members, can play an essential role in cancer drug efficacy and creating drug resistance.

To date, studies on animals have confirmed that the constitution of the gut microbiome can be severely affected by circulating androgen levels and castration [136,137], thus opening questions of whether the changes that the gut microbiome undergo during the therapies applied for the treatment of the PCa can further influence tumor progression and development. The ability of cancer therapies to affect and change the composition of the genitourinary microbiota to date has been studied in only a few studies.

Sfanos et al. profiled the composition of the gut microbiota in fecal samples of PCa patients with or without ADT or ATT and found a great diversity in gut microbiota composition among the PCa patients and with regard to other genitourinary disease patients [19]. In their study, the authors found that men taking oral ATT had a different genitourinary microbiota composition than men taking GNRH agonists/antagonists alone or men not undergoing ADT. The bacterial species capable of steroid biosynthesis (*Ruminococcaceae* and *Akkermansia muciniphila*) seems to be more abundant in the microbiota of men taking oral ATT. This may influence the progression following treatment and the immunotherapy. Notably, a recent study showed that ADT depletes androgen-utilizing *Corynebacterium* spp. in PCa patients and that the abiraterone acetate (inhibitor of androgen biosynthesis) treatment of CRPC patients remodels the gut microbiota, promoting the growth of anti-inflammatory gut commensal, *Akkermansia muciniphila*, and increasing microbial production of vitamin K2 [129]. In accordance, a further study pointed out that the gut bacteria can modulate the amount of circulating sex hormones by affecting host cells but also directly biotransforming or synthesizing them, driving the resistance to ADT [138]. The PCa patients showed an increased abundance of species, such as *Akkermansia muciniphila* and *Oscillospiraceae* (*Ruminoccocaceae*) spp., which were able to synthesize steroids hormones via CYP17A1-like bacterial enzyme(s). The treatment of cultured bacteria with CYP17A1 inhibitor abiraterone inhibits androgen synthesis. The manner in which the microbiota responds to reductions in androgen concentrations expanding the species that are able to synthesize androgens remains to be understood.

Quite recently, Pernigoni et al. demonstrated that surgical castration modifies the gut microbiota, increasing the growth of some bacterial species, such as *Ruminococcus gnavus* and *bacterioides acidifaciens*, in two mouse models of PCa, TRAMP-C1 and Ptenpc-/- [127]. The authors proposed that the depletion of the gut microbiota by antibiotics such as neomycin, ampicillin, vancomycin, and metronidazole may improve the outcome of ADT against PCa. Namely, it was postulated that ADT promotes the expansion of specific microbiota that is able to convert androgen precursors into active androgens, thus contributing to the onset of castration resistance in mice and, in that manner, annihilating the effects of ADT. It was observed that, in mice that had microbiota depleted by antibiotics, the changes in the circulating level of androgens as well as changes observed in the levels of expression of receptors led to a reduced growth of tumors, especially in CRPC animals. Furthermore, it was shown that the FMT with feces from wild-type mice did not induce changes in tumor growth and progression, while the feces from CRPC mice led to an increment in tumor growth. Additionally, the authors demonstrated that the administration of *Ruminococcus gnavus* to untreated mice increased the tumor growth. On the other hand, tumor growth was controlled by FMT from HSPC patients and *Prevotella stercorea* administration. Taken together, these results led the authors to conclude that, in CRPC, alternative sources of androgens can be provided by the gut microbiota, thus contributing to endocrine resistance. Nonetheless, how the microbiota responds to a reduction in androgen concentrations expanding the species that are able to synthesize androgens remains to be understood.

In another study aiming to investigate whether the fecal microbiota in PCa patients who had undergone prostatectomy or ADT differs, the fecal microbiota of 86 PCa-recruited patients (56 patients on ADT and 30 patients with prostatectomy) were analyzed by the 16S rRNA gene for α- and ß-diversities [139]. The authors observed a significantly lower alpha-diversity and *Firmicutes-to-Bacteroidetes* ratio in the ADT group. On the other hand, the biosynthesis of lipopolysaccharide and propanoate were enriched as well as the energy cycle pathways in the ADT group, thus offering a potential explanation for the metabolic complications in ADT. The changes in the gut microbiome by ADT and the relationship between testosterone levels and changes in the gut microbiota were investigated in another study involving 23 patients with PCa [140]. Bacterial 16 S rRNA gene-based microbiome and metabolome in fecal samples were analyzed. The α- and ß-diversities of gut microbiota decreased significantly at 24 weeks after ADT and significant positive correlation between the abundance of *Proteobacteria*, a known indicator of dysbiosis, and the concentration of lactate was observed. Moreover, the decline in testosterone levels resulted in detrimental changes in the gut microbiota.

Quite recently, Terrisse et al. investigated the relationship between ADT, gut microbiota, and thymus function in PCa [128]. The authors showed that microbiota manipulation (via cohousing or FMT or the administration of beneficial bacteria *A. muciniphila*) could influence the immune system and increase the efficacy of ADT in mice. In a group of PCa patients, the authors showed an enhanced, ADT-induced, thymic output in CRPC versus the HSPC. The presence of PCa alters the microbiota (in particular, by depletion of the beneficial bacteria *A. muciniphila*), and ADT reverses this effect. The ADT efficacy against PCa could be modulated by the immune system and the intestinal microflora; the restoration of a healthy gut microbiota may help to prolong tumor control by ADT, but the experiments with immunotherapies have failed to improve ADT efficacy [128].

PCa is an immune-responsive malignancy, but it remains unclear why it is less responsive to ICI than other tumors [141,142]. The efficacy of ICI may be related to the microbiota composition [143]. Currently, different human trials are investigating fecal transplants and ICI administration for immune-responsive malignancies. An ongoing clinical trial (NCT04116775) administrates the PD-1 antibody pembrolizumab to CRPC patients treated with enzalutamide in which the unresponsive patients were subjected to FMT.

A recent study by Peiffer et al., exploring the role of commensal genitourinary microbiota composition on tumor response to ICI, was performed to obtain the profile of the microbiota composition in a cohort of 23 patients with metastatic CRPC progressing in enzalutamide and treated with ICI pembrolizumab [144]. The authors investigated the α- and ß-diversities of the fecal samples collected before and after pembrolizumab treatment and calculated the integrated index reported as associated with checkpoint inhibitor response. Little differences were recorded in α- and ß-diversities between the responder and not-responder groups. In patients responding to pembrolizumab, the oral bacterium *Streptococcus salivarius* was increased, and the gut bacterium *Akkermansia muciniphila* was depleted. The integrated index showed no relationship with pembrolizumab treatment [144].

Fecal transplantation and/or bacterial modulation therapy have been suggested as the possible solution for PCa immunotherapy. Current evidence suggests FMT also as a strategy for the adverse event following ICI treatment.

Although several studies highlight the pivotal role of microbiota in PCa, substantial limitations hinder the development of new microbiome targets to potentially improve the efficacy of the established therapies.

## 7. Conclusions and Future Perspectives

Cancers are complex ‘ecosystems’ comprising many different cell types and non-cellular factors. PCa is a multifaceted heterogeneous disease associated with the acquisition of diverse hallmark capabilities: aberrant functioning of androgen receptor signaling, deregulation of vital cell physiological processes, inactivation of tumor-suppressive activity, disruption of prostate-gland-specific cellular homeostasis, and deregulated host–microbiome–tumor–viral interaction.

A growing body of studies shows an altered bacteria-rich environment in cancerous prostate tissues, unlike healthy prostate tissue. Through not fully understood, several studies reported chronic systemic inflammation and immune system modulation as the primary mechanisms by which the gut and the genitourinary microbiome enhance the prostate cancer risk. In addition, a relationship between the microbiome and therapy response has been demonstrated. However, only a few microbes directly cause cancer, while several promote host antitumor immunity. In this context, microbiome research remains controversial. There is a lack of data depicting host microbial species to their functional profiles and defining specific mechanisms by which microbes support cancer development and progression or affect treatment response.

Standardized sampling avoiding technical variables contamination, evaluation of multiple microbiomes (tissue, urine, blood, and feces), producibility on cancer microbiome data, and establishment of “gold-standard” pipelines may be able to address the contradictory findings and draw solid conclusions for microbially-based cancer diagnostics, prognostics, and microbial therapeutics.

Further research is needed to investigate in-depth the correlation between the gut and the genitourinary microbiome dysbiosis, chronic prostatic inflammation, and PCa insurgence and progression. The findings could pave the way for developing novel strategies for PCa prevention, novel treatment strategies, and better risk stratification tools.

## Figures and Tables

**Figure 1 ijms-24-01511-f001:**
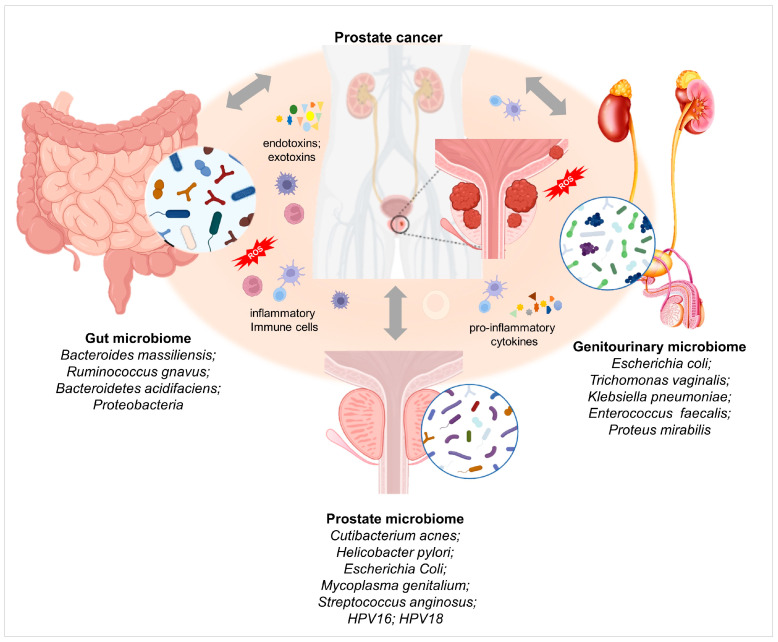
Gut, intraprostatic and genitourinary microbiomes and their association with prostate cancer (PCa) insurgence and progression. Image created using Biorender.com.

**Figure 2 ijms-24-01511-f002:**
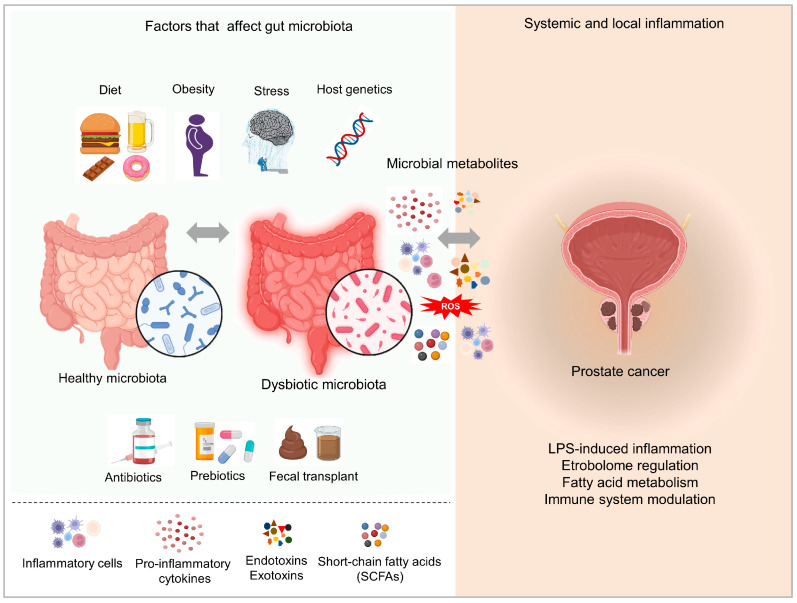
Factors affecting gut microbiota composition and mechanisms of action of the gut microbiome and the correlation with PCa. Image created using Biorender.com.

## Data Availability

Not applicable.

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
