# Peer review of "Microbiome and Prostate Cancer: A Novel Target for Prevention and Treatment"

_ijms, 2023, doi:10.3390/ijms24021511_

Round 1
Reviewer 1 Report
Dear authors,
After the review process, I have several comments: is necessary to add a table or a figure for a review, especially a figure that explains the relation between microbiota and PCa; also, the authors should explain the role of UTi, starting from the pathogenesis of rUTI and their effects on human urinary system homeostasis.
Best regards!
Author Response
We thank the reviewer for the comments and suggestions that can improve the review. We have followed the suggestions and included two figures. In addition, we also described the relationship of UTi and PCa. In the text named revised red you can see the added part in red.
Reviewer 2 Report
Line 20: “Second most frequent” author must tell in which population, because this big statement to mention or refer which
Line 63: which populations
Line 71: mention what are the types.
Line 73-74: how many confirmatory studies were performed?
Line 75: which tumor and by which mechanisms which you mention before in line 73-74
Line 105-107: This happens in the gut or in the Prostate, why is this limited to males not in females. Can the author explain this better
Line 114-115: What type of association was of E.coli and other enterobacteria with PCa
Line 121: They were causes or consequences for the PCa.
Line 167-168: If M1 and M2 both are abundant then what is the consequence e.g. inflammation increased or suppressed?
Line 171: why only in the prostate?
Line 381: why again prostate, why not on other organs?
Line 391: Explain the effects of HFD on sex hormones metabolism a bit more.
Line 394: I think author “oclusion” means “conclusion”
Line 431: Does this again cause or consequence?
Line 514: Explain which IAP family members were downregulated by Serratia marcescens
Author Response
We thank the reviewer for the comments and suggestions that can improve the review.
Line 20: “Second most frequent” author must tell in which population, because this big statement to mention or refer which
The statement refers to the male population worldwide ‘which remains the second most frequent male malignancy worldwide’. https://pubmed.ncbi.nlm.nih.gov/33538338/; https://www.wcrf.org/cancer-trends/worldwide-cancer-data/.
Line 63: which populations
Again, the statement refers to the population worldwide.
Line 71: mention what are the types.
This statement refers to the article Poor et al., as cited in the manuscript (Poore, G.D., et al., Microbiome analyses of blood and tissues suggest cancer diagnostic approach. Nature, 2020. 579(7800): p. 567-574 in which the authors have done extensive research about associations between cancer and microbiota across diverse cancer in a total of 33 cancer types, some of them adrenocortical carcinoma, acute myeloid leukemia, urothelial bladder carcinoma, brain lower grade glioma, breast invasive carcinoma, cervical squamous cell carcinoma, endocervical adenocarcinoma, colon adenocarcinoma, esophageal carcinoma ecc, including PCa. We added including PCa in the manuscript
Line 73-74: how many confirmatory studies were performed?
The comment personally is not very clear, since it is clearly referring to the reference Porter C.M., et al (Porter, C.M., et al., The microbiome in prostate inflammation and prostate cancer. Prostate Cancer Prostatic Dis, 2018. 21(3): p. 345-354) which is extensive almost meta-study from 2018 in which the authors are giving the extensive overview of the existing and relevant literature regarding the influence of the microbiome on various types of cancer.
In most cases, the cause of prostatic inflammation is unclear. Various potential sources exist for the initial inciting event, including direct infection, urine reflux-inducing chemical and physical trauma, dietary factors, oestrogens, or a combination of two or more of these factors. Furthermore, any of these could lead to a break in immune tolerance and the development of an autoimmune reaction to the prostate.
It has been found that microbes may influence the initiation and progression of PCa via direct interactions at the site of the cancer development, such as bacterial prostatitis, and indirect interactions, including the mediation of the immune response (Porter CM, et al., doi: 10.1038/s41391-018-0041-1).
In prostate cancer lesions, varied bacterial populations enhance the pro-inflammatory response, contributing to cancer development. For example, analysis of the microbial ecosystem of tumoral, peritumoral, and non-tumoral prostate tissue collected after radical prostatectomy found Enterobacteriaceae exclusively in tumoral samples (Cavarretta et al., doi: 10.1016/j.eururo.2017.03.029). Enterobacteriaceae is known to be more abundant in inflammatory and neoplastic conditions (Cavarretta et al., doi: 10.1016/j.eururo.2017.03.029.; Allen-Vercoe & Jobin, doi:10.1016/j.imlet.2014.05.014. De Marzo et al., doi: 10.1038/nrc2090.). In addition to contributing to prostatic inflammation, which is a risk factor for prostate cancer development, microbes have also been linked to response to treatment (De Marzo et al., doi: 10.1111/j.1365-2559.2011.04033.x.; Sfanos et al., doi: 10.1038/nrurol.2017.167).
Many different pathogenic organisms have been observed to infect and induce an inflammatory response in the prostate. These include sexually transmitted organisms, such as Neisseria gonorrhoeae, Trichomonas vaginalis and Treponema pallidum and those known to cause acute and chronic bacterial prostatitis, primarily Gram-negative organisms such as Escherichia coli. Although each of these pathogens has been identified in the prostate, the extent to which they cause prostate inflammation has yet to be established (De Marzo et al., doi: 10.1038/nrc2090.).
Indirect effect of microbiome-like immune modulation – it is already explained in the section above. Metabolic changes and epithelial damage can be related to urinary reflux. Chemical irritation from urine reflux has been proposed as an important factor for the development of chronic inflammation in the prostate (Kirby et al., doi: 10.1111/j.1464-410x.1982.tb13635.x.). Urine contains many chemical compounds that might be toxic to the prostate epithelium, with uric acid being particularly damaging (Persson & Ronquist PMID: 8583617). Crystalline uric acid directly engages the caspase-1-activating NALP3 (cyropyrin) inflammasome present in cells of the innate immune system (primarily macrophages), resulting in the production of inflammatory cytokines that can increase the influx of other inflammatory cells (Martinon et al., doi: 10.1038/nature04516.).
Another manner by which prostate inflammation might occur is the development of corpora amylacea (Drachenberg et al., PMID: 8964038). Namely it has been proposed that corpora amylacea contribute to prostate inflammation, persistent infection and prostate carcinogenesis because they are frequently observed adjacent to the damaged epithelium and focal inflammatory infiltrates.
Line 75: which tumor and by which mechanisms which you mention before in line 73-74
This sentence is refering to a reference number 12 (Ma, J., et al., Influence of Intratumor Microbiome on Clinical Outcome and Immune Processes in Prostate Cancer. Cancers (Basel), 2020. 12(9).) in which the authors perform extensive studyin which they measured the correlation of microbes with clinical variables including the Gleason score, TNM staging, and PSA values to observe associations between microbes and prostate cancer aggressiveness. Explaining the details of mechanisms by which certain microbioma can have anti-cancer effects and in which cases goes beyind the scope of this review.It is our believe that this explanation would need the section of its own which would be significant digresion from the principal thematic of this manuscript. If the reviewre perfers we can completely omit this sentence from the text.
Line 105-107: This happens in the gut or in the Prostate, why is this limited to males not in females. Can the author explain this better?
Here the statement refers to the pathogenic shift in the microbial species composition of both the gut and prostate.
To be more specific we changed the sentences into:
A pathogenic shift in the microbial species composition of the gut, intraprostatic, and genitourinary tract driving dysbiosis could lead directly or indirectly to local and/or systemic inflammation, an inflammatory state predisposing to loss of epithelial barrier integrity [11]. In turn, epithelial damage triggers an immune system response leading to the recruitment of the inflammatory cells, oxidative stress, and the consequent DNA damage, compensatory epithelial proliferation, establishing a feed-forward mechanism promoting prostatic intraepithelial neoplasia.
As for the issue of why this is limited to males, we think gender-specific differences in microbiota composition are a relevant point o be discussed, but not the focus of the review.
Males and females are known to have gender-specific differences in sex hormones, metabolism, gut microbiota composition, and immune systems may which may play a role in the sex differences in diseases.
How sex hormones, gender-dependent immunity differences, and gut microbiota composition impact the incidence and outcome of neoplastic diseases, is still unclear.
Line 114-115: What type of association was of E.coli and other enterobacteria with PCa
If I clearly understood the question, so if are directly or indirectly associated with the PCa, I would say both.
The microorganism can be involved in the process of prostate cancer by different biological mechanisms: for instance directly, by sustaining chronic inflammation and regulating the expression of various cytokines and chemokines in inflammatory cells, or directly by releasing metabolites leading to DNA oxidative damage, prostate cell genetic instability and epithelial cell proliferation that can directly drive prostate carcinogenesis.
Line 121: They were causes or consequences for the PCa.
They were causes.
Line 167-168: If M1 and M2 both are abundant then what is the consequence e.g. inflammation increased or suppressed?
In reference to your question, we have therefore modified the sentence as follows:
Malignant samples were also enriched for M1 and M2 macrophages as compared to benign tissue samples, suggesting an ongoing dysregulated inflammatory response.
Line 171: why only in the prostate?
The relationship of bacterial species present in specific tissues and their role in the carcinogenesis of that particular organ or gland requires further studies and investigation, and to date is unclear. Many factors may play a role in this multiphasic process.
Line 381: why again prostate, why not on other organs?
Many studies have found that chronic inflammation could play a major role as an etiologic factor in the development of several cancers, such as hepatocellular carcinoma, squamous cell carcinoma in the urinary bladder, colorectal cancer, and gastric cancer.
The causes of inflammation in the prostate can be various and among them may be decisive some bacteria related to prostatitis and sexually-transmitted disease, hormonal changes of estrogen, physical trauma caused, and urine reflux to the prostate gland.
Moreover, epidemiological studies have shown that obesity is associated with advanced prostate cancer and that obese men with prostate cancer have a poorer prognosis. HFD-induced prostate cancer progresses via adipose-secretory cytokines or chemokines. It has also been shown that both HFD and obesity in prostate cancer change the local profile of immune cells, in particular MDSC and macrophages.
In connection with these concepts, we decided to modify the sentence in this way:
Obesity resulting from a high-fat diet (HFD) induces chronic systemic inflammation via adipose-secretory cytokines or chemokines and may be involved in PCa progression through immune system-related mechanisms, particularly those associated with the function of MDSC and macrophages [79, 80].
Line 391: Explain the effects of HFD on sex hormones metabolism a bit more.
We thank the reviewer for his question so we can better elaborate and explain the relationship between HFD and sex hormone metabolism. Indeed, dietary intervention studies have not shown that a change in dietary fat and/or dietary fibre intake is associated with changes in circulating sex hormone concentrations over the short term, although there are some clinical and epidemiological evidence that diet is associated with circulating sex hormone levels in men.
Therefore, we prefer to change our sentence in this way:
What’s more,Moreover, HFD is able to significantly may impact the metabolism of sex hormones potentially leading to the progression of PCa. Nevertheless, current dietary intervention studies did not reveal that variations in dietary composition have any long-term major effects on circulating sex hormone levels in men.
Line 394: I think author “oclusion” means “conclusion”
Thanks to the reviewer for noticing the error. We then corrected with “conclusion”.
Line 431: Does this again cause or consequence?
In relation to this question, we were referring to the article of Liss et al. (ref. 102), who in their work showed a correlation between Streptococcus and Bacteroides species and PCa, but not the real mechanism and process that linked carcinogenesis and bacteria. However, they suggested that the fecal microbiome can metabolize and produce various products that may influence cancer through specific metabolic biosynthesis pathways.
Therefore, the sentence is revised as follows:
The study from Liss et al., showed that using 16S rRNA sequencing to detect rectal swabs, Streptococcus and Bacteroides species were higher in men with PCa than control subjects, and they speculated that gut bacteria may interfere through specific metabolic biosynthesis pathways on the risk of PCa.
Line 514: Explain which IAP family members were downregulated by Serratia marcescens
We thank the reviewer for this question so we can better specify which molecules are involved in this mechanism: the downregulation of the IAP family members relates to three molecules: XIAP, cIAP-1 and cIAP-2.
Therefore, the sentence is corrected as follows:
Serratia marcescens, (facultative anaerobes) can inhibit the growth of PCa cells through down-regulation of IAP family inhibitors including XIAP, cIAP-1 and cIAP-2 and activation of caspase-9 and caspase-3.
Round 2
Reviewer 1 Report
No other comments.